# A biomimetic redox flow battery based on flavin mononucleotide

Akihiro Orita[1,2], Michael G. Verde[1], Masanori Sakai[2] & Ying Shirley Meng[1]

The versatility in design of redox flow batteries makes them apt to efficiently store energy in large-scale applications at low cost. The discovery of inexpensive organic electroactive materials for use in aqueous flow battery electrolytes is highly attractive, but is thus far limited. Here we report on a flow battery using an aqueous electrolyte based on the sodium salt of flavin mononucleotide. Flavins are highly versatile electroactive molecules, which catalyse a multitude of redox reactions in biological systems. We use nicotinamide (vitamin B3) as a hydrotropic agent to enhance the water solubility of flavin mononucleotide. A redox flow battery using flavin mononucleotide negative and ferrocyanide positive electrolytes in strong base shows stable cycling performance, with over 99% capacity retention over the course of 100 cycles. We hypothesize that this is enabled due to the oxidized and reduced forms of FMN-Na being stabilized by resonance structures.

[1] Department of NanoEngineering, University of California, San Diego, 9500 Gilman Drive, La Jolla, California 92093, USA. [2] Core Technology Research & Innovation Center, Hitachi Chemical, 2200, Oka, Fukaya-shi, Saitama 369-0297, Japan. Correspondence and requests for materials should be addressed to Y.S.M. (email: shmeng@ucsd.edu).

Renewable energy power generation, such as from solar and wind, is highly attractive, but their intermittent nature is problematic. To compensate for this drawback, the use and development of appropriate energy-storage systems have been heavily pursued. Redox flow batteries (RFBs) have been attracting much attention for use as grid-scale storage because of unique advantages they present, such as their flexible, modular design and fast response time[1–4]. An RFB stores its energy using redox active materials dissolved in electrolyte, referred to as positive and negative electrolytes, which are separated by a membrane and circulated by pumps. While a number of RFBs use safe or inexpensive or high-energy density materials, combining all three important criteria into one has not yet been achieved. For example, vanadium RFBs have been the most successfully commercialized thus far, but the use of rare-metal ions, in vanadium, leads to quite expensive systems[5]. While less energy dense, RFBs using organic active materials have received increasing interest because of their promise to satisfy other important requirements such as low cost and sustainability. A variety of unique organic active materials have been reported for use in RFBs[3,6–13], a number of which have been inspired by research in Li-ion and Na-ion batteries[14–16]. The number of organic materials that can deliver stable cycling, however, has been very limited. The anthraquinone/bromide flow battery is one bright example, exhibiting stable performance over the course of 750 cycles[17,18]. Understanding and translating the beneficial properties contributing to quinone systems' stabilities should be a major goal in the development of alternative organic RFB systems.

Flavins act as a cofactor in many enzymes that catalyse a wide variety of biological reactions, and contain one of the most versatile *in vivo* redox centres[19,20]. The planar isoalloxazine ring forms the basic structure for riboflavin, flavin mononucleotide (FMN), and flavin adenine dinucleotide (FAD). The relationship of FMN-Na to the derivatives mentioned is illustrated in Supplementary Fig. 1a. The biosynthetic enzyme, flavokinase, catalyses the initial phosphorylation of riboflavin from adenosine triphosphate (ATP) to form FMN[21]. A fraction of FMN directly acts as a coenzyme. The large fraction of FMN combines with a second molecule of ATP to form FAD catalysed by FAD synthetase. Inspired by these biochemicals, flavins, such as riboflavin and lumichrome, have been demonstrated as solid-state electroactive materials for Li-metal batteries[22,23]. The concentration of flavins in those composite electrodes was no $>50$ wt%, however, due to the high content of conductive carbon necessary to compensate for their low electronic conductivity. In nature, flavins are often found dissolved in water, fat, or blood, such as in biological systems. As they are commonly more useful in this state, so may it also be in batteries. Interestingly, flavins have yet to be reported as an active material for any RFB, to the best of our knowledge.

In this study, we create a stable biomimetic RFB using a sodium salt of flavin mononucleotide (FMN-Na, Fig. 1a), known as riboflavin-5′-phosphate sodium salt, which serves as negative electrolyte. Figure 1b shows its schematic representation with potassium ferrocyanide, $K_4[Fe(CN)_6]$, as positive electrolyte, selected as one of many candidate redox couples (Fig. 1c), because of its stable reversibility under a strongly alkaline condition[13]. We henceforth refer to this RFB chemistry as FMN/Fe. Here we explore the use of FMN-Na in aqueous RFBs due to its relatively high water solubility compared with other flavins, such as riboflavin (vitamin B2) and lumiflavin. To increase its water solubility further, nicotinamide (NA), known as vitamin B3 (Supplementary Fig. 1b), was used as a biomolecular

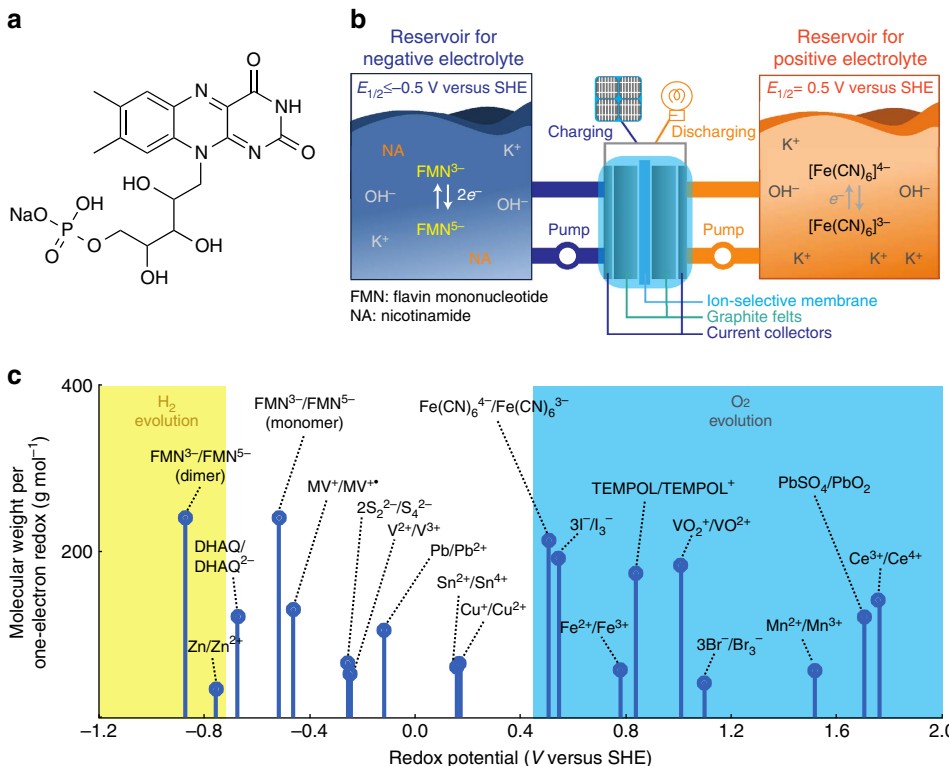

**Figure 1 | Cell schematic of an FMN/Fe battery.** (**a**) The molecular structure of riboflavin-5′-monophosphate sodium salt (FMN-Na, sodium salt of flavin mononucleotide). (**b**) Schematic representation of an RFB consisting of FMN-Na and ferrocyanide-based negative and positive electrolytes, respectively. (**c**) The redox potentials of various candidate redox couple highlighting regions of $H_2$ evolution (yellow) and $O_2$ evolution (blue) at pH 13. TEMPOL, 4-hydroxy-2,2,6,6-tetramethylpiperidin-1-oxyl; DHAQ, 2, 6-dihydroxyanthraquinone; MV, methyl-viologen.

**Figure 2 | Chemical structures of FMN-Na. (a)** FMN-Na at various protonation states of the phosphate group and third position nitrogen. (**b**) Two-electron redox reaction mechanism of FMN-Na in strongly alkaline conditions (pH > 10.2).

additive in FMN-Na electrolyte. NA, urea, and caffeine are known as hydrotropic agents, and their addition has been reported to enhance the water solubility of poorly water-soluble drugs, including flavins[24–28]. FMN/Fe RFBs with this design are shown to achieve stable cycling performance with a capacity retention of ∼99% after 100 cycles. We propose that the superior performance of FMN/Fe RFBs is enabled by resonance structures, which can stabilize oxidized and reduced forms of FMN-Na. This study provides a promising strategy to design energy storage system using organic active materials.

## Results

**Effects of pH on FMN-Na redox.** The electrochemistry of FMN-Na aqueous electrolyte was examined using cyclic voltammetry at various pH levels. The electronic structure of FMN has been previously studied in regard to biochemical applications, using ultraviolet–vis (ultraviolet–visible) spectroscopy and electron paramagnetic resonance[29–32]. Its electron-transfer mechanism in water has also been reported[33]. In an acidic aqueous solution (0.7 < pH < 6.2), the predominant species is FMN$^-$, whereby a proton from the phosphate group of FMN has been lost (Fig. 2a). In near-neutral and weakly basic solutions (6.2 < pH < 10.2), the phosphate group loses a second proton, forming FMN$^{2-}$. Furthermore, in a strong base (pH > 10.2), FMN can lose an additional proton from the nitrogen (N) located at the third position of the isoalloxazine ring to form FMN$^{3-}$. For reasons described in the following sections, we mainly explore the use of FMN in flow batteries under these strongly basic conditions. Figure 2b illustrates its redox reaction at pH > 10.2, whereby FMN$^{3-}$ undergoes a two-electron reduction to form FMN$^{5-}$. Due to the fact that FMN has been primarily studied in biological systems, and high pH conditions are uncommon in those environments, its electrochemistry in strongly basic conditions (pH > 12) has not yet been thoroughly investigated.

Figure 3a shows the cyclic voltammograms (CVs) using 10 mM FMN-Na at pH 5.5, 8.6, 10.0, and 13.0. At low pH (pH = 5.5), two peaks for both oxidation and reduction can be observed. This is indicative of a two-step redox mechanism, proceeding through the stable intermediate shown in Supplementary Fig. 2a. At pH 8.6 and 10.0, only one redox peak is observed, centred at ∼ − 0.5 V versus Ag/AgCl, which suggests that the two-electron reaction proceeds in a concerted manner[30–32]. Furthermore, this

result shows that FMN$^{2-}$ is stable in this pH range, as the CVs do not change from 8.6 to 10.0. In general, the redox potential of reactions involving proton exchange shifts as a function of pH, based on the Nernst equation: $\Delta E = -m/n\ 0.059 \times$ pH at a temperature of 298 K, where $\Delta E$ is the shift from the standard redox potential, and m ($mH^+$) and n ($ne^-$) are the number of protons and electrons involved in the redox reaction, respectively. Since no voltage shift occurs in the pH range of 8.6–10.0, this indicates that the redox reaction of FMN$^{2-}$ does not directly involve proton exchange, as shown by the mechanism provided in Supplementary Fig. 2b.

At pH 13.0, a significant cathodic shift in the redox potential is observed, which can be explained by the fact that FMN$^{3-}$ is less likely reduced, due to its large negative charge compared with FMN$^{2-}$ or FMN$^-$. The oxidation and reduction peak separation at pH 13.0 is 48 mV, which is larger than the theoretical value expected for a reversible two-electron reaction (28.5 mV). This suggests that the FMN-Na redox reaction can be described as a quasi-reversible reaction. Nevertheless, the peak separation and current density at pH 13.0 were much improved compared with lower pH. From these results, the strongly alkaline condition is shown to be preferable for FMN-Na used as negative electrode. This concept along with the detailed kinetics of FMN$^{3-}$ is more thoroughly described in later sections of this paper.

**Dimerization of FMN-Na.** In alkaline conditions, FMN is known to exhibit photoinduced electron transfer[33] and undergo hydrolysis with OH$^-$ at high temperature (353 K)[34,35]. For this reason, we aimed to explore the temporal stability of the FMN-Na redox reaction. Figure 3b shows the CVs of FMN-Na in 1 M KOH aqueous electrolyte (pH 13.0), up to 100 h after its preparation. The initial redox couple was centred at − 0.730 V versus Ag/AgCl. Over time, however, the peak intensity at this location decreased, while a redox couple at more negative potential developed. The developing reduction peak was located at − 1.15 V, while its reversible oxidation peak was more broadly located between − 0.85 and − 1.2 V versus Ag/AgCl. It should be noted that the temporal change in CVs could be observed regardless of the presence or absence of fluorescent light or oxygen gas in the electrolyte (controlled by N$_2$ purging).

To first examine the possibility that hydrolysis of FMN-Na caused a change in the CVs over time, the CV of 10 mM FMN-Na

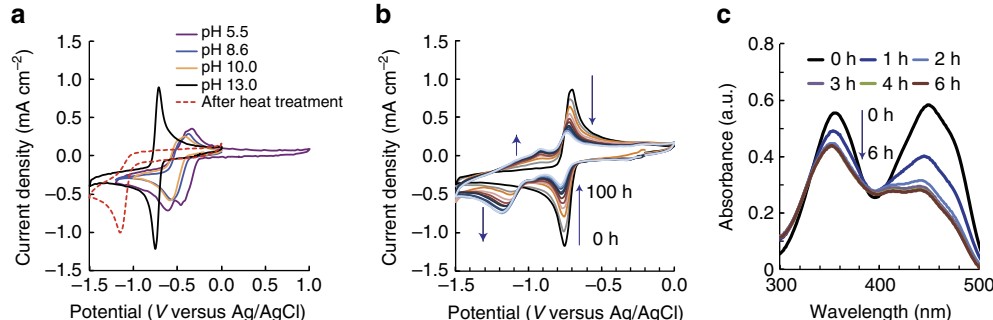

**Figure 3 | Cyclic voltammograms and ultraviolet–vis spectra of FMN-Na electrolytes.** (**a**) CVs of 10 mM FMN-Na and 1 M KCl (pH 5.5), 10 mM FMN-Na and 1 M KCl in pH 9.1 buffer (pH 8.6), 10 mM FMN-Na and 1 M KCl in pH 10 buffer (pH 10.0), and 10 mM FMN-Na and 1 M KOH (pH 13.0) aqueous solutions; the dashed CV represents the latter electrolyte after treatment at 363 K for 2 h. (**b**) CVs of 10 mM FMN-Na 1 M KOH aqueous electrolyte at 10 h intervals over the course of 100 h (sweep rate: 10 mV s$^{-1}$). (**c**) Ultraviolet–vis spectra of 50 µM FMN-Na in 1 M KOH aqueous electrolyte 0–6 h after electrolyte preparation.

in 1 M KOH aqueous electrolyte was measured after heating to 363 K. High temperatures are known to decompose FMN by the hydrolysis reaction shown in Supplementary Fig. 3 (ref. 35). As shown by the CV in Fig. 3a (dashed), the hydrolysed product exhibits only a negative current, indicating an irreversible redox reaction. The new peak formed over time at room temperature, however, is reversible (Fig. 3b). We conclude, therefore, that hydrolysis does not contribute to the formation of the second peak, and is likely negligible near room temperature. The reversibility of this second peak is further demonstrated in the full-cell testing section described later.

We also performed temporal ultraviolet-vis measurements using 50 µM FMN-Na in 1 M KOH aqueous electrolyte up to 6 h after electrolyte preparation (Fig. 3c). The absorption peaks observed at 375 and 450 nm, which can be assigned to $\pi \rightarrow \pi^{\star}$ transitions[36], decreased over time, up to 3 h. A factor that may contribute to the evolving redox reaction and the decrease in absorption over time may be due to the dimerization of FMN-Na. The observed decrease in ultraviolet-vis absorption was not due to the alkaline hydrolysis of FMN-Na because the decrease could be observed even at pH 5.5 and 8.6 (Supplementary Fig. 4). On the basis of NMR analysis, FMN has been known to stack with itself in aqueous solution, to form dimers[33,37,38]. Furthermore, the extinct coefficient of its dimer is lower than that of its monomer at pH 7 (ref. 39). Its formation could, therefore, result in the decreased absorbance measured. The dipole–dipole interaction of stacked monomers generally lowers its energy states[37], leading to higher reduction energies, which could also explain the cathodic potential shift observed. The fact that CV measurements were shown to continuously change up to >100 h, while the ultraviolet-vis spectra showed no change after 3 h, may be due to the differences in concentration of those electrolytes (10 mM versus 50 µM FMN-Na, respectively). The equilibrium constant of dimer formation has been reported to depend on the concentration of monomer at pH 7 (ref. 39), and our future studies aim to examine this equilibrium in strong base.

**Kinetics of FMN-Na.** To expand upon the kinetics of FMN-Na, its diffusion coefficient and kinetic rate constant were investigated using rotating disk electrode (RDE) at a pH of 13.0. Figure 4a illustrates the limiting diffusion current at various rotating angular velocities of the electrode, ranging from 200 to 3,000 r.p.m., using electrolyte immediately following preparation. The half-wave potential ($E_{1/2}$) of FMN-Na was −0.726 V versus Ag/AgCl (−0.517 V versus SHE) at a pH of 13. Figure 4b displays the dependence of the limiting current on rotating

velocity. The fact that the linear trend line crosses the origin means that there was no observed chemical reaction that preceded or followed the redox reaction of FMN-Na electrolytes in the time frame used within the experiment. The diffusion coefficient of FMN-Na was determined to be $(1.3 \pm 0.1) \times 10^{-6}$ cm$^2$ s$^{-1}$ using the Levich equation[40]. Figure 4c shows Koutecký–Levich plots, which provide the heterogeneous rate constant $i_k$ at each potential. $i_k$ represents the current in the absence of any mass-transport effects, and can be observed for only quasi-reversible and irreversible reactions[41]. These values were used to form the Tafel plot shown in Fig. 4d, which exhibits good linearization. From this plot, the transfer coefficient for the reduction of FMN-Na was calculated to be $\alpha = 0.50$. This value indicates that the energy barriers for oxidation and reduction of FMN-Na are symmetric[42]. The kinetic rate constant for the reduction of FMN-Na was determined to be $k_0 = (5.3 \pm 0.5) \times 10^{-3}$ cm$^2$ s$^{-1}$, which is greater than those for inorganic redox couples, such as $V^{3+}/V^{2+}$ ($5.3 \times 10^{-4}$ cm s$^{-1}$) (ref. 43) and $VO^{2+}/VO_2^+$ ($2.8 \times 10^{-6}$ cm s$^{-1}$) (ref. 43) (Supplementary Table 1). Future studies aim to explore the effect of dimerization on those kinetic parameters.

**Electrochemistry of an FMN/Fe redox flow battery.** Initial FMN/Fe RFBs were tested using 0.06 M FMN-Na and 0.1 M $K_4[Fe(CN)_6]$ in 1 M KOH at a current density of 5–20 mA cm$^{-2}$. The CV and RDE measurements using 20 mM $K_4[Fe(CN)_6]$ and 1 M KOH aqueous positive electrolyte are shown in Supplementary Fig. 5. The excess amount of FMN-Na was used because the purity of FMN-Na was >93% based on anhydrate and easily hydrated. Figure 5a illustrates the charge–discharge curves at a current density of 5, 10, and 20 mA cm$^{-2}$. The initial discharge capacity at 10 mA cm$^{-2}$ was 1.31 Ah l$^{-1}$, which is close to the theoretical capacity of 1.34 Ah l$^{-1}$. The open-circuit voltage (OCV) at 100% SOC and the average discharge voltage at a current density of 10 mA cm$^{-2}$ were 1.40 V and 1.03 V, respectively. The charging curve shows two plateaus around 1.15 V and 1.50 V, which we hypothesize results from reduction of the monomer and dimer of FMN-Na, respectively. Interestingly, the discharge curve does not mirror the two discrete plateaus observed on charge curve. The oxidation peak of the dimer measured using CV (Fig. 3a) was also quite broad. In addition, the high-voltage plateau became less distinct with higher current densities, which resulted in lower charge capacities overall. This phenomenon may be explained by the lower kinetic rate of dimer FMN-Na than that of monomer FMN-Na. Figure 5b,c show the charge–discharge profiles,

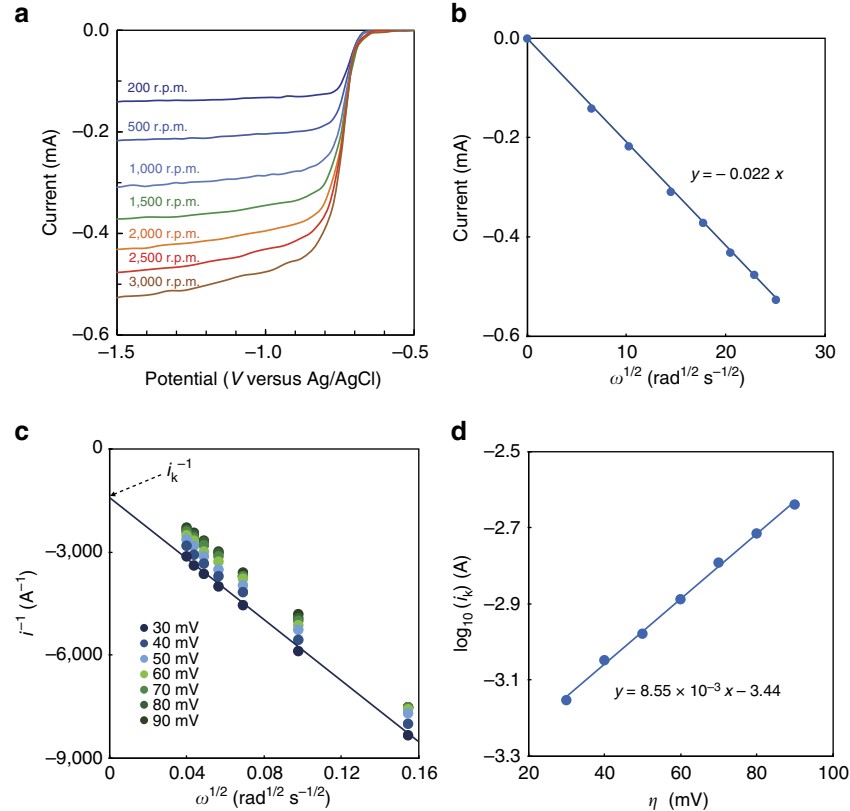

**Figure 4 | Electrochemistry of an FMN-Na electrolyte.** (**a**) RDE measurements at rotating electrode speeds from 200 to 3,000 r.p.m. using 10 mM FMN-Na in 1 M KOH aqueous solution (pH = 13.0). (**b**) The limiting current ($i$) versus the square root of the rotation velocity (Levich-plot). (**c**) Koutecký–Levich plot. (**d**) Tafel plot. $\eta$ is the difference between the measured potential and the formal redox potential.

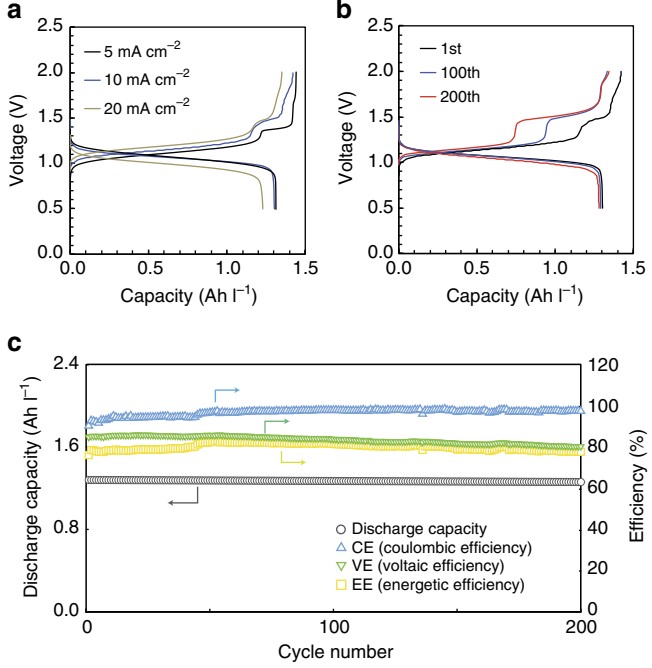

**Figure 5 | Electrochemistry of an FMN/Fe RFB at a low concentration.**
0.1 M $K_4[Fe(CN)_6]$ in 1 M KOH aqueous positive electrolyte and 0.06 M FMN-Na in 1 M KOH aqueous negative electrolyte were used. (**a**) Charge–discharge profiles at a current density of 5, 10 and 20 mA cm$^{-2}$. (**b**) Charge–discharge profiles at 1st, 100th, and 200th cycles at 10 mA cm$^{-2}$. (**c**) Cycling discharge capacity and efficiencies of an RFB at a current density of 10 mA cm$^{-2}$.

discharge capacities and efficiencies over a number of cycles. Highly stable discharge capacities are observed over the course of 200 cycles, which took 124 h. The Coulombic efficiency improves slightly over time, which is associated with the significant extension of the high voltage plateau due to dimer reduction. Voltage efficiency slightly decreased on cycling due to the increasing proportion of dimer reduction, which takes place at slightly higher voltages.

We hypothesize that the stable cyclability is in part enabled by the resonance structures of oxidized and reduced forms of FMN-Na, as shown in Supplementary Fig. 6 (refs 44,45). A variety of resonance structures lead to electron delocalization, which lowers the overall potential energy of FMN-Na, thereby making it less prone to decomposition. The general ability of resonance states to stabilize organic redox species can be adapted to explain the durability of other compounds, such as those that are quinone-based, as well. The resonance structures of anthraquinone, which was stably cycled in the quinone/Br system for 750 cycles[18], are shown in Supplementary Fig. 7a. On the contrary, the worse cycling ability of benzoquinone[9] compared with anthraquinone may be explained by its relatively poor diversity of resonance structures at the reduced state (Supplementary Fig. 7b). From these trends, we propose that the stabilization of redox pairs by resonance structures can be a promising strategy to design energy storage systems based on organic active materials.

**Energy-density enhancement.** The full-cell performance of FMN/Fe RFB was investigated at a higher concentration to assess the limits of its energy density and cycling performance under ideal working conditions. The maximum water solubility of $K_4[Fe(CN)_6]$ in 1 M KOH aqueous solutions at 298 K was

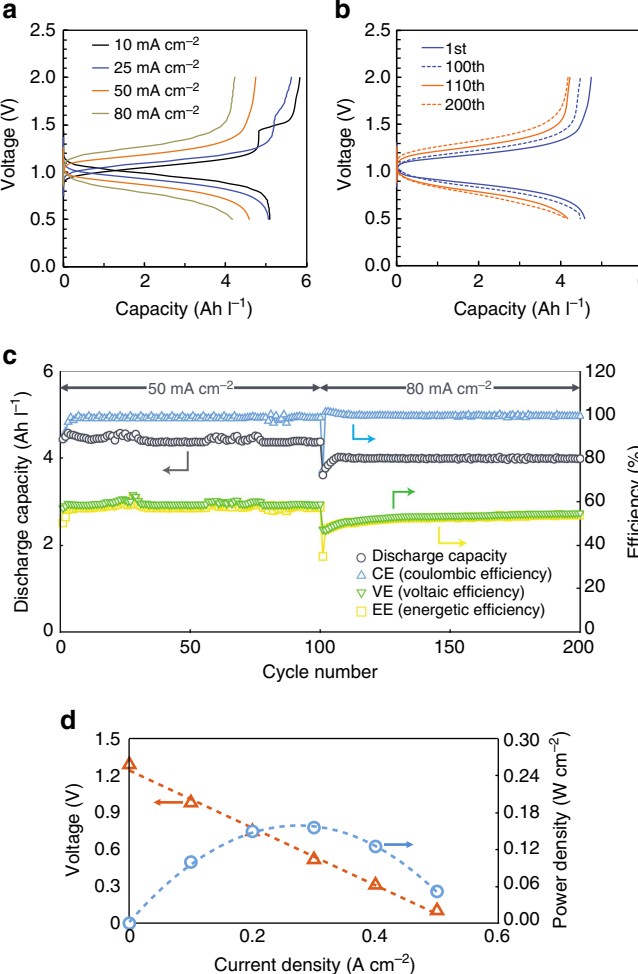

**Figure 6 | Electrochemical performance of an FMN/Fe RFB at a high concentration.** 0.4 M $K_4[Fe(CN)_6]$ in 1 M KOH aqueous positive electrolyte and 0.24 M FMN-Na and 1 M NA in 1 M KOH aqueous negative electrolyte were used. (**a**) Charge–discharge profiles at 10, 25, 50 and 80 mA cm$^{-2}$. (**b**) Charge–discharge curves at 1st and 100th cycles at 50 mA cm$^{-2}$, and 110th and 200th cycles at 80 mA cm$^{-2}$. (**c**) Cycling discharge capacities and efficiencies. (**d**) Voltage- and power-density dependencies on discharge current density.

$\sim 0.5$ M. As highlighted in Supplementary Fig. 8a, the water solubilities of FMN-Na in 1 M $H_2SO_4$ (pH 0.8), 1 M KCl (pH 5.5) and 1 M KOH aqueous solutions (pH 13.0) were determined to be $\sim 10$, 50 and 100 mM, respectively. The highest water solubility in strongly alkaline conditions may be explained by the fact that FMN-Na has higher polarity than in acidic or neutral solution due to the large negative charge ($FMN^{3-}$), which results in the most stable solvated state by water. To further enhance the water solubility of FMN-Na, NA (nicotinamide) was added as a hydrotropic agent[24]. FMN-Na showed a maximum solubility of $\sim 1.5$ M in an aqueous solution of 1.0 M KOH by adding 3 M NA (Supplementary Fig. 8b). The viscosity of this electrolyte increased to 3.2 mPa s, compared with the 0.06 M FMN-Na 1 M KOH solution of 1.2 mPa s. Determining the optimum concentrations of FMN-Na will be explored in the future studies. Supplementary Fig. 9a shows the CV of 10 mM FMN-Na with 10 mM NA in 1 M KOH. Compared with the CV in Fig. 3a, the peak potentials and shape do not change, indicating that the addition of NA has no effect on the redox reaction.

Figure 6a illustrates charge–discharge profiles of an RFB with 0.4 M $K_4[Fe(CN)_6]$ in 1 M KOH positive electrolyte and 0.24 M

FMN-Na and 1 M NA in 1 M KOH negative electrolyte at a current density of 10–80 mA cm$^{-2}$. The two charge plateaus assigned to reductions of monomer and dimer FMN-Na can be observed only at a low current density (10–25 mA cm$^{-2}$). The CV of 0.24 M FMN-Na electrolyte also shows no distinct peak for the redox reaction of dimer at high sweep rates (Supplementary Fig. 9b,c). The initial capacity measured at a current density of 25 mA cm$^{-2}$ was 5.03 Ah l$^{-1}$, which was close to the theoretical capacity of 5.36 Ah l$^{-1}$. The OCV at 100% SOC and the average discharge voltage of the initial cycle at 25 mA cm$^{-2}$ were 1.30 V and 0.96 V, respectively, which delivered an energy density of 4.83 Wh l$^{-1}$.

Figure 6b,c shows the cycling discharge capacities, efficiencies, and charge–discharge curves over 200 cycles, which took 76 h, at a current density of 50 and 80 mA cm$^{-2}$. The discharge capacity retention after 100 cycles was 99%, and the Coulombic efficiency was $>99\%$ at a current density of 80 mA cm$^{-2}$. Their stabilities are comparable to the quinone system, and greater than other systems using organic active materials (Supplementary Table 2). In addition, no FMN-Na was detected in the positive electrolyte after 200 cycles (Supplementary Fig. 10), which suggests that crossover is not an issue, likely due to the negatively charged $FMN^{3-}$ and $FMN^{5-}$. The observed low capacity (1% during 100 cycles) may be due to the alkaline hydrolysis even at room temperature or poor reversibility of dimer FMN-N. A hydrotropic agent has been reported to the ability to suppress the base-catalysed hydrolysis of riboflavin[46], and our future studies aim for further improvement. To evaluate the power density of the RFB, a polarization curve was plotted, which is shown in Fig. 6d. The peak power density was determined to be 0.16 W cm$^{-2}$ at a current density of 0.3 A cm$^{-2}$, which is greater than the all-vanadium RFB (0.12 W cm$^{-2}$ at 0.15 A cm$^{-2}$). Currently, the full-cell energy density is limited by the solubility of the positive electrolyte. For one-electron process, the concentration of positive electrolyte should be 3 M to make use of FMN's maximum solubility of 1.5 M FMN-Na (with 3 M NA). This negative electrolyte concentration will enable a full cell with high energy and power densities. We are currently exploring new active materials as high water solubility alternatives to $K_4[Fe(CN)_6]$.

## Discussion

FMN-Na is composed of earth-abundant elements and is environmentally friendly. This work demonstrates that FMN-Na is a promising active material for sustainable RFBs. We propose that a variety of resonance structures of its reduced state enable stable redox cyclability. The future design and incorporation of resonance-stabilized active materials may be beneficial to the development of long-life energy storage systems. We have also shown that the hydrotropic agent nicotinamide, known as vitamin B3, demonstrates the ability to increase the solubility of FMN-Na. FMN-Na has a solubility $>1.5$ M with 3 M NA (0.1 M without), and the FMN-Na solution can achieve the high capacity density of 81 Ah l$^{-1}$. Although materials with higher water solubility than $K_4[Fe(CN)_6]$ as positive electrolyte must be further explored to test the limits of FMN as a negative electrolyte, this study reveals its promise for use in RFBs.

Quinonic and flavin compounds, such as vitamins, FMN and nicotinamide adenine nucleotide (NAD), have been naturally selected for use in biochemical systems through the long evolutionary process. Thus, the biomimetic energy storage system described in this work is derived from nature's wisdom. Not only FMN-Na, but also other biochemicals found in cells, the citric acid cycle, enzymes and coenzymes, may inspire the design of sustainable energy storage systems. Recently, metal complexes based on biochemicals have been also proposed for RFBs[3].

Titanium complexes using catechol, pyrogallate, lactate and citrate, which can be found in plants and animals, have shown a reversible redox reaction[47,48]. Strategies inspired by biological systems, such as the utilization of hydrotropic agents and biochemicals, may expedite the successful exploration of sustainable ecofriendly batteries.

## Methods

**Materials.** All chemicals were used as received without further purification. Riboflavin-5′-phosphate sodium salt (FMN-Na, purity ≥93% based on the anhydrate substance) was purchased from TCI America. 96% $H_2SO_4$, Nafion212, potassium ferrocyanide, potassium hydroxide, potassium chloride, nicotinamide and boric acid were obtained from Sigma-Aldrich. The pH 9.1 buffer solution was prepared by adding 100 mM boric acid and 50 mM KOH into deionized water. The pH 10 buffer solution was purchased from Fisher Chemical. The pH levels of some solutions were adjusted using potassium hydroxide, pH 9.1 buffer, and pH10 buffer solutions. The pH value of each electrolyte was measured with an Oakton pH meter (pH-2700). The water solubility was examined at 288 K.

**Full-cell performance.** Full-cell tests were carried out using an Arbin BT-2000 battery tester (Arbin Instrument) and RFB cell (Scribner Associates) comprised of poly(tetrafluoroethylene) (PTFE) frame, graphite plates with flow fields, anodized aluminum current collector, and graphite felt electrodes (GFA6, SGL) with an active area of 25 cm². Galvanostatic evaluations were conducted at currents from 10 to 0.5 A cm⁻² between 0 and 2.0 V. Nafion 212 was used as an ion-exchange membrane after a treatment for exchanging $H^+$ with $K^+$. This exchange procedure was carried out in 1 M KOH at 80 °C for 6 h, followed by being washed with deionized water, according to previous literature[49]. Forty millilitre of positive and negative electrolyte was pumped at a flow rate of 40 ml min⁻¹ through a peristaltic pump. Coulombic efficiency, voltaic efficiency and energetic efficiency were calculated by the ratios of the average discharge capacity to the average charge capacity, the average discharge voltage to the average charge voltage and the average discharge energy to the average charge energy, respectively. Two FMN/Fe RFB compositions were primarily explored: the first was 0.1 M $K_4[Fe(CN)_6]$ and 1.0 M KOH aqueous positive electrolyte and 0.06 M FMN-Na in 1.0 M KOH aqueous negative electrolyte; the second was 0.4 M $K_4[Fe(CN)_6]$ and 1.0 M KOH aqueous positive electrolyte and 0.24 M FMN-Na and 1 M nicotinamide in 1.0 M KOH aqueous negative electrolyte. An OCV was measured after the initial charge process at 10 or 20 mA cm⁻². The capacity density was calculated based on the total volume of positive electrolyte (40 ml) and negative electrolyte (40 ml). Energy density was calculated by multiplying the discharge capacity and the average discharge voltage of the initial cycle.

**Voltammetry.** CV and RDE voltammetry were conducted using an Arbin BT-2000 battery tester (Arbin Instrument) and a BASi RDE-2. A BASi Ag/AgCl aqueous reference electrode (RE-5B, MF-2079, 3 M NaCl filling solution), platinum wire auxiliary counter electrode (MW-1033, 0.5 mm diameter) and glassy carbon working electrode (MF-2066, 3 mm diameter) was used. CV curves were measured at a sweep rate of 10 mV s⁻¹.

For RDE experiments, current was measured from −0.2 to −1.5 V versus Ag/AgCl at a 5 mV s⁻¹ sweep rate, during which time the electrode was rotated at 200–3,000 r.p.m. The half-wave potential $E_{1/2}$ was calculated as the voltage, which showed the half of the limiting current, at 200 r.p.m. The limiting currents measured at −1.5 V were plotted versus the rotation rate ($\omega$). The slope was fit using the Levich equation[40]: $i = 0.620 \ nFAcD^{2/3}v^{-1/6}\omega^{1/2}$, where $i$ is the limiting current, $n$ is the number of electrons in the reaction ($n = 2$ for FMN-Na), $F$ is the Faraday constant (96485 C mol⁻¹), $A$ is the surface area of the working electrode (0.0707 cm²), $c$ is molar concentration in mol cm⁻³, $v$ is the kinetic viscosity in cm² s⁻¹ ($9.0 \times 10^{-3}$ cm² s⁻¹ for 10 mM FMN-Na and 1 M KOH aqueous solution) and $\omega$ is the routing angular velocity in rad s⁻¹. The heterogeneous rate constant was calculated from the vertical-axis intercepts at each potential using the Koutecký–Levich equation[40]: $1/i = 1/i_k + 1/0.620 \ nFAcD^{2/3}v^{-1/6}\omega^{1/2}$. It should be noted that the Koutecký–Levich equation can only be used for quasi-reversible and irreversible reactions[41]. $\log_{10}(i_k)$ was plotted versus $\eta$, which was defined by the difference between the measured potential and the formal redox potential $E^{0'}$. Here the half-wave potential at 200 r.p.m. was used as the formal redox potential $E^{0'}$. The fitted line was expressed by the Tafel equation: $\log_{10}(i) = \log_{10}(nFcAk_0) + \alpha nF\eta/2.303RT$, where $n$ is the number of electrons in the rate limiting step ($n = 1$ for FMN-Na)[50], $\alpha$ is the transfer coefficient, $R$ is the universal gas constant (8.314 J K⁻¹ mol⁻¹), $T$ is temperature, $k_0$ (cm s⁻¹) is the standard kinetic rate constant, and the concentrations of redox couples were assumed to be the same.

**Ultraviolet–vis spectroscopy.** Ultraviolet–vis spectra were measured using a UV-VIS spectrometer UV-1800 (Shimadzu Scientific Instrument) and quartz spectrophotometer cells (Aldrich, 10 mm optical path length). The spectrum was measured at a wavelength range of 300–800 nm at a sampling interval of 0.5 nm.

**Data availability.** The data that supports the findings of this study are available from the corresponding author on request.

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

## Acknowledgements

UCSD team gratefully acknowledges the funding from Advanced Research Projects Agency—Energy (ARPA-E) under Grant No. DE-AR0000520. A.O. and Y.S.M. acknowledge the partial funding from Hitachi Chemical.

## Author contributions

A.O. conceived and conducted the research, and A.O. and M.G.V. designed the experiments. All authors discussed and analysed the results. A.O., M.G.V. and Y.S.M. wrote the manuscript.

## Additional information

**Competing financial interests**: The authors declare no competing financial interests.

**How to cite this article**: Orita, A. *et al.* A biomimetic redox flow battery based on flavin mononucleotide. *Nat. Commun.* **7,** 13230 doi: 10.1038/ncomms13230 (2016).

