## [Peer review file · Nature Communications]

Reviewers' comments:

Reviewer #1 (Remarks to the Author):

This submission describes a new electro-active material for aqueous-based redox flow batteries. The stabilization of the anion and dianion through the incorporation of heterocyclic nitrogen atoms (replacing carbon atoms in anthraquinone) is posed by the authors to be a major reason for high solubility and stability. Performance of this material in $\text{KFe}(\text{CN})_6$ flow cells shows minimal loss in capacity with cycling and is able to be charged at high current densities. The researchers bring a biologically-inspired molecule to the electrochemical energy storage community for an improved approach for aqueous RFBs. The methods for analysis are appropriate and thorough, with some details missing as described below.

The anthraquinone/bromide flow battery is one bright example, exhibiting stable performance over the course of 750 cycles^{16,17}.

Could the authors comment on the viscosity of the higher-concentration FMN-Na solutions and if changes as a function of pH were observed?

For FMN-Na, which is described at 93% purity, what impurities were present? I.e. the phosphonic acid form (protonated instead of sodium salt) or something else?

For the flow battery experiments, the total time to complete 200 cycles should be added to the text.

Given that membrane cross-over was not observed, could the authors comment on possible mechanisms for capacity loss evident in Figure 6b?

Given similarities in structure to the quinones with $\text{KFe}(\text{CN})_6$ reported by Aziz and cited here, I suggest the following citations be included: "Alkaline quinone flow battery" DOI: 10.1126/science.aab3033

Change "UV-VIS" to "UV-vis".

Reviewer #2 (Remarks to the Author):

This is a well-written paper that describes the use of a flavin for RFBs. The flavins, like quinones, are found extensively in nature. After characterizing key electrochemical properties of the flavin at different pH, the authors demonstrate their performance in RFBs. While the results are very interesting, there are several issues that should be addressed including, but not limited to:

- Considering strict definition, use of term biomimetic is questionable
- There has been substantial progress made with metal complexes that is not reflected in the manuscript.
- Details of the CVs are difficult to see, and therefore difficult to interpret; authors indicate one redox couple at pH 10, the CV does not appear to be consistent. Should use coulometry to definitively argue two electron process with one peak.
- Significant mechanistic details are placed in the supplementary material
- There is significant degradation of the active species at pH 13 as indicated in figure 3. It appears that features in CVs after cycling for 100 h and after heat treatment are somewhat similar; authors conclude that they are not. Not convinced that loss of current density is due to dimerization.
- The authors should consider uncertainly in results, in particular when comparing results with those in the literature. For example, kinetics for VIII/VII are likely within experimental error of

those for FMN-Na. Also the kinetics don't account for possibility of dimerization.

- This would appear to be a 1 V cell. Voltage limits for the RFB experiments appear to exceed stability limits for various active species.
- Given kinetics of system, lack of plateaus at higher current densities is surprising
- Some broad conclusions regarding application of the chemistry in commercial RFBs might need to be toned down in lieu of what appears to be instability/reactivity of the flavin. At high concentrations the problems will likely get worse.

Once addressed this paper might be suitable for publication, however, the question will be whether this use of this particular species is sufficiently novel. In some regards, this is like changing the ligand on a metal complex and reporting the results. Are there fundamental differences between the flavins and other biological redox species that makes them particularly attractive?

Reviewer #3 (Remarks to the Author):

In the submitted article, the authors researched on the biomimetic redox flow battery based on vitamin derivatives which is a topic that has been recently highlighted with the increasing importance of large-scale and sustainable energy storage systems due to their potential structural/chemical diversity/flexibility, non-toxicity, and low-cost. Therefore, I recommend this article to be published in Nature communications after a major revision. Before publication, typo errors need to be carefully checked such as "Highly stable discharge capacity is observe over" on page 7.

C1. I would like to know how you could suggest that the only one redox peak is related to two-electron reaction at pHs 8.6 and 10.0. It would be helpful for readers if there is an evidence for two-electron reaction of those pHs.

C2. It would be great if you can suggest the direct evidence of FMN-Na dimerization. UV-vis spectroscopy is not enough to be an evidence of dimerization. IR spectroscopy or NEXAFS spectroscopy can be useful tools for verifying dimerization.

C3. In comparison with Fig. 5 (a) and Fig. 5 (b), same condition, but different voltage profiles are observed. Even though they have same condition for charge/discharge, the 2nd charge plateaus are different in both side. I would like you to explain about these two ambiguous phenomena.

C4. The authors mentioned that 0.06M FMN-Na was applied to full cell test, but the molarity ratio of catholyte and anolyte is not 1:2. It would be helpful for readers if you explain why.

C5. What is the main cause that hydrotropic agents can elevate the solubility of FMN-Na? I would like to know what range of materials can be elevated of their solubility by hydrotropic agents, and what the requirements of selecting hydrotropic agents.

Response to Reviewer #1 Comments

Comment 1-1

“Could the authors comment on the viscosity of the higher-concentration FMN-Na solutions and if changes as a function of pH were observed? ☐”

Response 1-1

Yes, the concentration of FMN-Na affects the viscosity of its solution. However, 3 M of nicotinamide (NA) has to be added to obtain the maximum solubility of FMN-Na, NA also affects its viscosity.

In addition, because of the low water-solubility of FMN-Na in an acid, we can compare the viscosities of FMN-Na solutions only at low FMN-Na concentration. At the 10mM of FMN-Na, the viscosities at various pHs depended on only the kind of a supporting electrolyte.

For these reasons, we added the effect of FMN-Na concentration on the viscosity only under an alkaline condition at Page 9 Line 7, as follows;

“FMN-Na showed a maximum solubility of *ca.* 1.5 M in a 1.0 M KOH aqueous solution by adding 3 M NA (Supplementary Fig. 8b), even though its viscosity was also increased to 3.2 mPa s compared to a 0.06 M FMN-Na 1 M KOH solution (1.2 mPa s). The optimum concentration of FMN-Na has to be explored in the future study.”

Comment 1-2

“For FMN-Na, which is described at 93% purity, what impurities were present? I.e. the phosphonic acid form (protonated instead of sodium salt) or something else?”

Response 1-2

This 93% is based on the anhydrated state. We assume that impurities consist of hydrated water and bisphosphate. We addressed that 93% was based on the anhydrated substance in the experimental section (Page 12 Line 2), as follows;

"All chemicals were used as received without further purification. Riboflavin-5'-phosphate sodium salt (FMN-Na, purity \geq 93% based on the anhydrate substance) was purchased from TCI America."

Comment 1-3

For the flow battery experiments, the total time to complete 200 cycles should be added to the text.

Response 1-3

We added the sentences at Page 9 Line 24, as follows;

"Highly stable discharge capacity is observed over the course of 200 cycles, which took 124 h."

and Page 8 Line 4 as follows;

"Fig. 6b and Fig. 6c show the cycling discharge capacities, efficiencies, and charge-discharge curves over 200 cycles, which took 76 h, at a current density of 50 and 80 mA cm⁻²."

Comment 1-4

"Given that membrane cross-over was not observed, could the authors comment on possible mechanisms for capacity loss evident in Figure 6b?"

Response 1-4

We are now exploring the capacity loss mechanism, so we added the sentences at Page 10 Line 4, as follows;

"The observed capacity loss (1% during 100 cycles) may be due to the alkaline hydrolysis even at room temperature or poor reversibility of dimer FMN-Na, and our future studies aim to clarify a capacity loss mechanism."

Comment 1-5

"Given similarities in structure to the quinones with KFe(CN)₆ reported by Aziz and cited here, I suggest the following citations be included: "Alkaline quinone flow battery" DOI: 10.1126/science.aab3033."

Response 1-5

We agree your suggestion. We added the citation as ref. 13 and the sentence at Page 2 Line 26, as follows;

"Fig. 1b shows its schematic representation with potassium ferrocyanide, K₄[Fe(CN)₆], as positive electrolyte, selected as one of many candidate redox couples (Fig. 1c), because of its stable reversibility under an strongly alkaline condition¹³."

Comment 1-6

"Change "UV-VIS" to "UV-vis."

Response 1-6

As you suggested, UV-vis is correct, so we have changed to UV-vis.

Response to Reviewer #2 Comments

Comment 2-1

“Considering strict definition, use of term biomimetic is questionable”

Response 2-1

A “biomimetic” means an art and science that mimic biological systems. In our study, not only an active material but also an additive to enhance the water solubility were selected as bio-inspired materials. The mechanism for the enhancement of water solubility is unclear yet, but we could conceive this idea by imitating a biological system. In this meaning, we used the term “biomimetic”.

We think that there are many editors whose specialty is biochemistry in Nature Communications, so we can consult about the manuscript title with editors.

Comment 2-2

“There has been substantial progress made with metal complexes that is not reflected in the manuscript.”

Response 2-2

Because this study focuses on the bio-inspired materials for flow batteries, we added the sentences and references related to bio-related metal complexes for flow batteries at Page 11 Line 13, as follows;

“Recently, metal complexes based on biochemicals have been also proposed for RFBs³. Titanium complexes using catechol, pyrogallate, lactate, and citrate, which can be found in plants and animals, have shown a reversible redox reaction^{45,46}.”

Comment 2-3

“Details of the CVs are difficult to see, and therefore difficult to interpret; authors indicate one redox couple at pH 10, the CV does not appear to be consistent. Should use coulometry to definitively argue two electron process with one peak.”

Response 2-3

The redox reactions mechanism of FMN at pH 3-11 has been well researched in many literatures. To help readers understand that our mechanism was based on those literatures, we put the reference numbers on the corresponding sentence at Page 4 Line 16, as follows;

“At pHs 8.6 and 10.0, only one redox peak is observed, centered at *ca.* -0.5 V vs. Ag/AgCl, which suggests that the two-electron reaction proceeds in a concerted fashion²⁹⁻³¹”

At pH 13, it is evident that the redox reaction of FMN-Na goes through two-electron reaction from the result that the peak separation in CV was 48 mV, which was low than theoretical value for one-electron redox reaction (57.5 mV).

Comment 2-4

“Significant mechanistic details are placed in the supplementary material”

Response 2-4

Fig.3c (UV-vis spectra) in the revised manuscript was moved from the supplementary Figure. The other figures in the supplementary material partly contain already reported results, so these are remained in the supplementary file.

Comment 2-5

“There is significant degradation of the active species at pH 13 as indicated in figure 3. It appears that features in CVs after cycling for 100 h and after heat treatment are somewhat similar; authors conclude that they are not. Not convinced that loss of current density is due to dimerization.”

Response 2-5

The charged capacity observed below 1.2 V, which could be assigned to monomer FMN-Na, was lower than the discharge capacity. This means that dimer FMN-Na, which could be charged above 1.3 V, contributed to the discharge capacity. In addition, the new broad oxidation peak was observed in CVs after 100h, whereas CV after heat treatment showed the only reduction peak.

So we concluded that temporal change of FMN-Na is not direct degradation. We think that the loss of capacity at high current density may be due to the low kinetic of dimer FMN-Na.

Comment 2-6

“The authors should consider uncertainly in results, in particular when comparing results with those in the literature. For example, kinetics for VIII/VII are likely within experimental error of those for FMN-Na. Also the kinetics don't account for possibility of dimerization”

Response 2-6

The significant figure of a surface area of an electrode used in CVs and RDE measurements is 10% according to the RDE manufacturer. In addition, we measured the kinetic rate constant 3 times, and calculated value fell in this range. So we changed the those values with 10% error at Page 7 Line 3, as follows;

“The diffusion coefficient of FMN-Na was determined to be $1.3 \pm 0.1 \times 10^{-6} \text{ cm}^2 \text{ s}^{-1}$ using the Levich equation³⁹. ”

and at Page 7 Line 11, as follows;

“The kinetic rate constant for the reduction of FMN-Na was determined to be $k_0 = 5.8 \pm 0.6 \times 10^{-3} \text{ cm}^2 \text{ s}^{-1}$, which is greater than those for inorganic redox couples”

In addition, the kinetic rate constants of vanadium ions and FMN-Na were different by more than one order of magnitude. Even after considering the experimental error, we can conclude that FMN shows the larger kinetic rate constant than vanadium ions.

Comment 2-7

“This would appear to be a 1 V cell. Voltage limits for the RFB experiments appear to exceed stability limits for various active species.”

Response 2-7

The redox potentials of dimer FMN-Na and $K_4[Fe(CN)_6]$ are around -1.2 V, and 0.5 V vs Ag/AgCl, respectively. By considering the overpotential, the limit voltage for charging should be set to be around 2 V. It should be noted here that a carbon electrode could broaden the electrochemical window of water.

Comment 2-8

“Given kinetics of system, lack of plateaus at higher current densities is surprising.”

Response 2-8

The lack of plateaus at high current density consists the broadening of the second oxidation peak, assigned to dimer FMN-Na, in CVs at a high sweep rate (supplementary Fig. 9b). This may be due to the lower kinetic rate of dimer FMN-Na than that of monomer. We added the sentence (yellow-highlighted) at Page 7 Line 25, as follows;

“The oxidation peak of the dimer measured using CV (Fig. 3a) was also quite broad. In addition, the high voltage plateau became less distinct with higher current densities, which resulted in lower charge capacities overall. This phenomenon may be explained by that lower kinetic rate of dimer FMN-Na than that of monomer FMN-Na.”

Comment 2-9

“Some broad conclusions regarding application of the chemistry in commercial RFBs might need to be toned down in lieu of what appears to be instability/reactivity of the flavin. At high concentrations the problems will likely get worse. Once addressed this paper might be suitable for publication, however, the question will be whether this use of this particular species is sufficiently novel. In some regards, this is like changing the ligand on a metal complex and reporting the results. Are there fundamental differences between the flavins and other biological redox species that makes them particularly attractive?”

Response 2-9

Flavin mononucleotide (FMN) has never reported as an active material for any kind of batteries, such as lithium-ion, sodium-ion, and flow batteries. We would like to claim that this material is sufficiently novel. Even though some metal complexes have been reported as active materials, there are few complexes have been based on the biological design. We added those examples as addressed at Response 2-2.

This study can also provide a strategy to select a good active material by utilizing a variety of resonance structures at both oxidized and reduced states that act to stabilize it and prevent decomposition or side reaction.

We used the relatively low concentration of FMN-Na to match with the low water solubility of ferrocyanide and to suppress the volume change between positive and

negative electrolytes. We believe that the use of high concentration of FMN-Na does not cause worse degradation, because the high concentration of FMN-Na suppresses the dimerization (Fig. 3b (10 mM) vs. Supplementary Fig. 9b (240mM)). The only problem at high concentration may be an increase in an electrolyte viscosity.

There are many biological redox species, such as NAD, FAD, purine, pyrimidine, and pyridine nucleotide, etc. Though we concluded that flavins was suitable for RFBs in the points of its reversibility and relatively high water solubility; however, there is still much room to examine those biochemical for RFBs, including as a ligand.

Response to Reviewer #3 Comments

Comment 3-1

“I would like to know how you could suggest that the only one redox peak is related to two-electron reaction at pHs 8.6 and 10.0. It would be helpful for readers if there is an evidence for two-electron reaction of those pHs.”

Response 3-1

The mechanisms redox reactions of FMN at pH 3-11 have been well researched by other people, as shown in the introduction section. To help readers understand that our mechanism was based on literatures, we put the reference numbers on the corresponding sentence at Page 4 Line 16, as follows;

“At pHs 8.6 and 10.0, only one redox peak is observed, centered at *ca.* -0.5 V vs. Ag/AgCl, which suggests that the two-electron reaction proceeds in a concerted fashion²⁹⁻³¹”

At pH 13, it is evident that the redox reaction of FMN-Na goes through two-electron reaction from the result that the peak separation in CV was 48 mV, which was low than theoretical value for one-electron redox reaction (57.5 mV).

Comment 3-2

“It would be great if you can suggest the direct evidence of FMN-Na dimerization. UV-vis spectroscopy is not enough to be an evidence of dimerization. IR spectroscopy or NEXAFS spectroscopy can be useful tools for verifying dimerization.”

Response 3-2

We measured the IR spectroscopy to obtain an evidence of dimerization according to your informative suggestion. The following Figure shows the IR spectra of an electrolyte measured 0 h and 50 h after its preparation. There was no distinct difference between those spectra. It may be difficult to detect π - π interaction by using FT-IR.

Figure | FR-IR spectra of (a) 1 M KOH and (b,c) 0.1 M FMN-Na in 1 M KOH aqueous solutions. Spectra in (a) and (b) were measured immediately after electrolyte preparation. Spectrum in (c) was measured 50 h after electrolyte preparation. Peak assignment is based on ref. 1. FT-IR spectroscopy was performed on a Nicolet 6700 FT-IR spectrometer (ThermoFisher Scientific) equipped with iD7 ATR accessory and with a spectral resolution of 4 cm^{-1} . A spectral range of 4000–800 cm^{-1} was recorded.

We may measure NEXAFS and STXM at a synchrotron if we can secure a beam-line time, and we think it should be another different paper from this manuscript.

Comment 3-3

In comparison with Fig. 5 (a) and Fig. 5 (b), same condition, but different voltage profiles are observed. Even though they have same condition for charge/discharge, the 2nd charge plateaus are different in both side. I would like you to explain about these two ambiguous phenomena.

Response 3-3

The cell with different graphite felts were used in Fig. 5(a) and 5(b), so we unified the same condition including felts in this manuscript. We apologize for confusing you. We are now elaborating on the effect of a cell structure and components, such as felts.

Comment 3-4

“The authors mentioned that 0.06M FMN-Na was applied to full cell test, but the molarity ratio of catholyte and anolyte is not 1:2. It would be helpful for readers if you explain why.”

Response 3-4

The purity of FMN-Na was 93% based on the anhydrated state according to a manufacturer report. We used the excess amount of FMN-Na by considering the absorbed water or impurities.

We added the sentences at Page 7 Line 16 and Page 12 Line 3, as follows;

“The excess amount of FMN-Na was used because the purity of FMN-Na was more than 93% based on anhydrate and easily hydrated.”

“All chemicals were used as received without further purification. Riboflavin-5'-phosphate sodium salt (FMN-Na, purity \geq 93% based on the anhydrate substance) was purchased from TCI America.”

Comment 3-5

“What is the main cause that hydrotropic agents can elevate the solubility of FMN-Na? I would like to know what range of materials can be elevated of their solubility by hydrotropic agents, and what the requirements of selecting hydrotropic agents. “

Response 3-5

It is unclear even for us. We have selected the hydrotropic agent to enhance the water solubility of FMN-Na because it has been already reported by literatures in the pharmaceutical field; however, it should be noted that this manuscript firstly reported the strong effect of nicotinamide for flow batteries.

As you know, it is difficult to predict the solubility of compounds in various solvents. There are few indicators, such as solubility parameters, so we need to examine the solubility of each composition in trial and error.

REVIEWERS' COMMENTS:

Reviewer #1 (Remarks to the Author):

The revisions to the manuscript satisfactorily address my questions and comments. I recommend this manuscript for publication without revision.

Also, regarding reviewer comment 3-2, I agree that IR spectroscopy is not useful for pi-pi dimers and that UV-vis analysis was appropriate. Perhaps in the future, the authors might explore the near IR region, acquiring spectra from 300-3300 nm, to see if changes are evident in lower energy regions. To be clear, this is not a suggestion for the present manuscript.

Reviewer #3 (Remarks to the Author):

All comments were satisfactorily answered in the revised manuscript, therefore, I recommend this article to be published in Nature communications.